# Robotic Care Equipment Improves Communication between Care Recipient and Caregiver in a Nursing Home as Revealed by Gaze Analysis: A Case Study

**DOI:** 10.3390/ijerph21030250

**Published:** 2024-02-22

**Authors:** Tatsuya Yoshimi, Kenji Kato, Keita Aimoto, Izumi Kondo

**Affiliations:** 1Laboratory for Clinical Evaluation with Robotics, Assistive Robot Center, National Center for Geriatrics and Gerontology, Obu 474-8511, Japan; yoshimit@ncgg.go.jp; 2Department of Rehabilitation Medicine, National Center for Geriatrics and Gerontology, Obu 474-8511, Japan; 3Assistive Robot Center, National Center for Geriatrics and Gerontology, Obu 474-8511, Japan

**Keywords:** gaze analysis, gerontechnology, nursing homes, quality of care

## Abstract

The use of robotic nursing care equipment is an important option for solving the shortage of nursing care personnel, but the effects of its introduction have not been fully quantified. Hence, we aimed to verify that face-to-face care is still provided by caregivers in transfer situations when using robotic nursing care equipment. This study was conducted at a nursing home where the bed-release assist robot “Resyone Plus” is installed on a long-term basis. Caregiver gaze was analyzed quantitatively for one user of the equipment during transfer situations, and communication time, which involved looking at the face of the care recipient, as well as face-to-face vocalization, was measured. The caregiver spent 7.9 times longer looking at the face of and talking to the care recipient when using Resyone than when performing a manual transfer. In addition, the recipient was observed to smile during Resyone separation, which takes about 30 s. The results indicate a possible improvement in the QOL of care recipients through the use of robotic nursing care equipment as a personal care intervention. The ongoing development of robot technology is thus expected to continue to reduce the burden of caregiving as well as to improve the QOL of care recipients.

## 1. Introduction

Addressing the issue of a declining and aging population, and especially the decrease in the working age population, has become urgent and important not only in Japan but also worldwide [1]. Although the Ministry of Health, Labour, and Welfare of Japan has taken the lead in developing the use of robot technology for nursing care [2], it has been reported that there is a low adoption rate of various types of robotic nursing care equipment, and that about half of the facilities that have introduced such equipment do not seem to be making effective use of it [3]. However, we have shown that the use of some robot technologies might be able to reduce the burden of caregivers during transfers, for example, and should improve the efficiency of caregiving tasks [4]. In addition, we have demonstrated the effects and changes that can occur to care recipients when caregivers have more time for caregiving tasks as a result of the reduced burden. For example, we found that long-term use of transfer support equipment increased communication between caregiver and care recipient, as evidenced by an increase in the number of times caregivers spoke to the patient [5] and an increase in the number of smiles of both individuals [6]. One expected result is a positive effect on the quality of life (QOL) of the cared-for person. Here, we measured communication time by analyzing a caregiver’s eye gaze when talking to the care recipient and looking at the care recipient’s face (eye contact), the latter of which is considered to be an important aspect of the body language of caregiving [7]. Furthermore, it has been found that nurses also mainly use gaze, nods, and smiles toward patients to improve communication [8].

In this study, we analyzed the caregiver’s gaze (a specific point at which the caregiver is looking) during transfer assistance (bed separation scenario) using Resyone Plus [9,10] (abbreviated as Resyone; Panasonic Age-Free Corporation, Osaka, Japan). Resyone can assist a bedridden subject to leave the bed without needing a transfer, because half of the electric nursing care bed separates to become a wheelchair (Figure 1). In particular, we demonstrated here that as a practical personal care intervention, assistance using robotic nursing care equipment not only reduces caregiver burden but also has the potential to improve the QOL of care recipients.

## 2. Materials and Methods

### 2.1. Equipment

Resyone Plus is a motorized care bed in which half of the bed can separate and be used as a wheelchair [10]. This equipment does not operate autonomously. However, just as assist suits are included in the category of nursing care robots, here, we define Resyone Plus as a bed separation support robot, because it assists subjects to get out of bed.

### 2.2. Intervention Period

The period over which this entire study was conducted has been described previously [6]. The study commenced 13 months after the introduction of this equipment to a nursing home. Briefly, nine months after the introduction of Resyone to a nursing home, we made a “proposal to expand the living area” using the robot, to the extent that it did not interfere with their nursing care duties. Four months later, this study was carried out.

### 2.3. Subjects

The study was conducted in a nursing home for the elderly with one care recipient and one caregiver. Care recipient: female (75–79 years age range, 150–160 cm range, 40–49 kg range); level of care required, 5; level of daily living independence for disabled elderly, C; level of daily living independence for elderly with dementia, IV. The care recipient was unable to speak but was sometimes able to respond vocally with groaning sounds when spoken to by the caregiver. We refer to these exchanges between caregiver and the care recipient as ‘vocalization’. The care recipient was selected because she had been using the Resyone equipment continuously for 13 months. Caregiver: male (30–34 years age range, 160–170 cm range, 60–70 kg range); nationally certified caregiver. The caregiver was selected because he was primarily responsible for the chosen care recipient. Subject for simulation of transfer assistance operation: male (35–39 years old, 170–180 cm, 50–60 kg); physical therapist.

### 2.4. Gaze Analysis

Gaze analysis was undertaken using a spectacle-type device (Pupil Invisible, Pupil Labs GmbH, Berlin, Germany), as worn by the caregiver in Figure 1. From the recorded video, we visually checked in how many frames (30 frames/s) the gaze (green circle) was directed towards the face and to other points and measured the number of seconds the gaze was directed towards each point. The gaze path (a 2 s period) is indicated by a pink line (Figure 2). To document vocalization, the frames in the video in which voices were present were counted using video editing software (Adobe Premiere Elements 2021). The scene in which the subject was vocalizing was visually confirmed, and the number of frames counted. Gaze and vocalization were recorded simultaneously, so the work time was increased over the actual operation for each vocalization. Since the duration of each vocalization was less than 0.4 s (<12 frames), which is short compared to the duration of gaze distribution, it did not have a significant impact on the overall analysis time. The analysis showed two occasions when a smile was evident on the care recipient’s face during the Resyone separation phase, as shown in Appendix A.

### 2.5. Transfer Phases during Resyone Separation

We classified the phases of transfer using Resyone as follows (Figure 3): A_Voice call/Moving blanket, B_Moving the back cushion, C_Moving the leg cushion, D_Lateral movement, E_Handrail up, F_Resyone Separation, G_Handrail up, H_Putting on socks and shoes, I_Resyone pillow insert, J_Raising the backrest, and K_Grooming/Blanketing.

### 2.6. A Simulation of Manual Transfer

To confirm that there is a marked difference in the gaze distribution compared to manual transfers, we performed a simulation of a transfer that caregivers manually perform without Resyone (Figure 4). In the simulation, we asked a therapist to ‘help’ a care practice doll to leave the bed (i.e., to get the doll up and transfer it to a chair), and we performed gaze analysis. The care practice doll weighed 27 kg and was 157 cm tall.

### 2.7. Communication Time

Communication time was defined as the time spent vocalizing with the care recipient and looking at the care recipient’s face, either momentarily or with a full gaze.

### 2.8. Ethics Approval and Consent to Participate

The study protocols have been reviewed and approved by the Ethics and Conflict of Interest Committee of the National Center for Geriatrics and Gerontology (acceptance no. 1552). All participants in the study provided informed consent. The care recipient was unable to speak or sign, so her husband signed the informed consent form on her behalf. All methods were carried out in accordance with relevant guidelines and regulations.

## 3. Results

### 3.1. A Typical Gaze Pattern during Transfer

An example of actual gaze analysis and a list of points to which the gaze was directed are shown in Figure 2. When the caregiver is gazing at the face, e.g., in panel 03 of Figure 2, this is distinguished by the absence of eye movement, and a momentary glance at the face is distinguished by the presence of eye movement (revealed by a pink line). However, over the whole of the episode recorded in Figure 2, no eye contact with the care recipient was ever observed. A possible reason for this was that, depending on the specifications of the device, the viewpoint may shift if the user himself/herself does not calibrate the gaze when wearing the device. Due to the limitations of time, in this case, calibration was not performed by the caregiver himself, but rather by one of the authors, which may mean that eye contact was not registered by the device, even if it was actually made.

### 3.2. Result of Gaze Analysis (Using Resyone)

Figure 3 shows the results of eye gaze analysis when a caregiver assisted the patient to leave the bed using Resyone. Speech, facial gazing, and a momentary glance are highlighted in red and pink colors. In total, the period of gaze analysis lasted 245 s, which is a relatively long time for transfer assistance, and for each of the phases A_Speech, F_Resyone Separation, and K_Grooming, it can be seen that there was a great deal of facial gazing. During these periods, it was observed that the care recipient smiled twice (panels A and B, Appendix A), especially during the F_Resyone Separation phase.

### 3.3. Gaze Analysis of Manual Transfer (Rising and Transfering)

For comparison, a simulation of a manual transfer was performed using a care practice doll. The phases of the transfer are shown in Figure 4A (a to h), and the results of the gaze analysis are shown in Figure 4B. The therapist talked to the doll, lifted it up, and then continued with the transfer. The entire process of rising from the bed and subsequently transferring the doll to a chair took 21.4 s. Even in such simulations, it is important that the therapist spends sufficient time (nearly 2 s) looking at the doll’s face before each movement, such as phase C_Placing the hands under the neck and phase F_Placing the hands under the armpits.

### 3.4. Communication Time

As shown in Figure 5, there was a 7.9-fold difference between manual rise and transfer assistance (7.63 s) and when assisted by Resyone (60.4 s). This confirmed that the amount of communication time between caregiver and care recipient increased significantly with Resyone.

## 4. Discussion

It has been reported in recent years that the burden of caregiving is becoming heavier not only in Japan but worldwide [11,12,13,14]. In such a situation, the use of robotic technology has been recommended [15,16,17]. However, its effectiveness has not been fully verified so far. The use of robots in nursing care is often thought to reduce the quality of care, but this is not necessarily true.

In a previous paper, we reported an increase in positive facial expressions of care recipients as a result of long-term use of Resyone [6]. In this study, we measured quantitative changes in communication time using eye gaze analysis and video analysis. We have shown in another paper that the use of a transfer support robot called Hug has also been reported to increase normal conversation [5]. However, we feel that the observation of an increase in communication time between caregiver and care recipient resulting from the use of robotic care equipment is novel.

When we suggested in our previous study that Resyone might be used to expand the living area [6], the caregivers involved indicated that there was no significant increase in the burden of caregiving, and they were able to manage their caregiving activities without increasing the number of transfers. Such care operations, which included visits after meals to new places in the care home, should also be applicable to other facilities.

From the above series of studies, we found that when robotic nursing care equipment is used over the long term for bed release or transfers, the time spent communicating with the caregiver increases, although the number of bed releases did not increase as expected. This suggests that similar caregiving operations with other transfer support devices could be organized to increase communication without increasing the caregiving burden. Although the use of a robot takes more time when assisting transfers, it can expand the range of activities available to the care recipient and, as shown in this study, increase communication time, which indicates the usefulness of the equipment. Prior to this study, communication was evaluated by speech analysis or time studies [4,5], but these are not parameters of direct communication. In contrast, the gaze analysis in this study confirmed that the caregiver was looking directly at the patient, and that this led to the expression of a smile in the care recipient, a result that shows the importance of direct and personal communication by caregivers.

Regarding research on caregiver eye contact and gaze, there are several studies on whether it is possible to gain trust from children [18] and patients [8,19,20] or measuring the degree of stress among professional nurses [21,22]. But there are few investigations on communication between caregiver and the recipient involving face gaze. This study is one of the few examples in which the influence of the caregiver’s gaze on the care recipient is visualized. We have observed smiles on the subject’s face, but this was not a quantitative result, nor did it show better QOL. However, the current study covers a single type of robotic equipment and a limited number of subjects. It is necessary to consider facial expression analysis, etc. for further research. Furthermore, no comparative device with Resyone has been commercialized, so we could not provide true control data in this case.

As previously reported, by using the Resyone continuously for several months, caregivers became accustomed to using it, and as a result, the burden of caregiving was reduced [4,6]. At that stage, we suggested that the subject be taken to the entrance or corridor, and the caregiver was able to do so without difficulty. The care operation was able to change because of the sustainable use of the device, and as a result, facial expression of the subject was improved [6]. For facility residents who are almost bedridden and require intensive nursing care, increased communication is expected to improve QOL and contribute to the “realization of wellbeing” advocated by the WHO [23], as well as to promote “social participation of the elderly” in the future as described in a literature [24]. This is a good example of how even a single-function care robot can have such effects when used in a sustainable manner. We believe that this type of assisting device can be implemented for personal care and can be applied to home care as well.

## 5. Conclusions

This study indicates that the use of robotic nursing care equipment may improve the QOL of care recipients, which is considered to be an effective use of a piece of equipment. The ongoing development of robot technology is thus expected to reduce the burden of caregiving and to improve the QOL of care recipients as well as the quality of caregiving. That may be true not only in nursing homes, but also in private homes.

## Figures and Tables

**Figure 1 ijerph-21-00250-f001:**
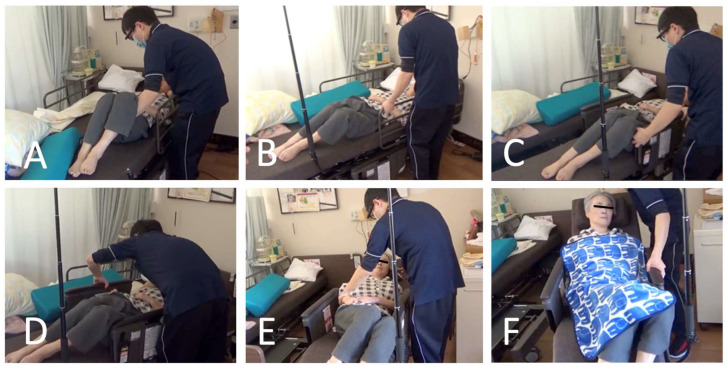
Steps for rise assistance, separating half of the Resyone for use as a wheelchair. The steps are lateral movement on the bed (**A**), removal of the handrail of the Resyone bed (**B**), wheelchair separation from the bed (**C**), raising up the handrail (**D**), raising the back of the wheelchair (**E**), and moving as a wheelchair (**F**).

**Figure 2 ijerph-21-00250-f002:**
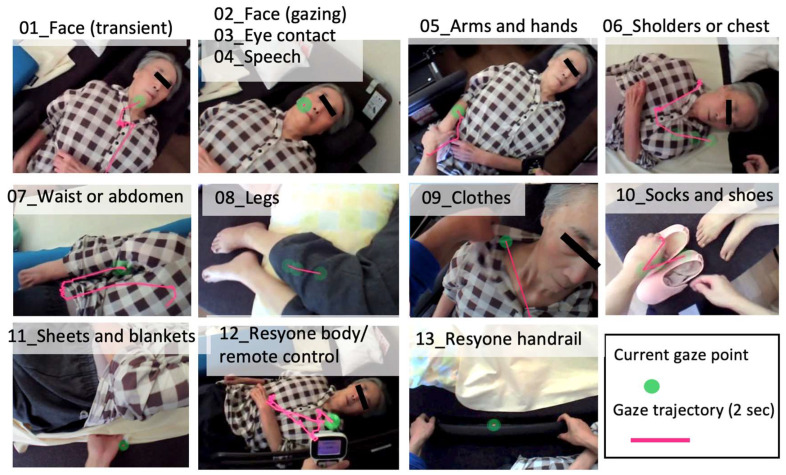
Gaze targets during the transfer using Resyone. The current viewpoint is indicated by a yellow-green circle, and the eye movements during the preceding 2 s are represented by a pink line. The gaze target and image, which was cropped from each analysis, are shown in the figure as 01_Face (transient), 02_Face (gazing), 03_Eye contact, 04_Speech, 05_Arms and hands, 06_Shoulders or chest, 07_Waist or abdomen, 08_Legs, 09_Clothes, 10_Socks and shoes, 11_Sheets or blankets, 12_Resyone body/remote control, and 13_Resyone handrail. As the images obtained from analyses 02 to 04 were almost the same, one image is shown.

**Figure 3 ijerph-21-00250-f003:**
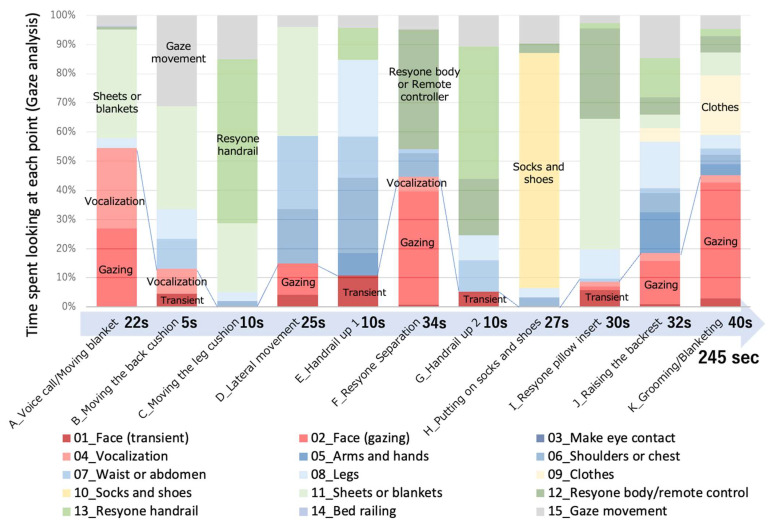
Gaze analysis of caregiver during transfer assistance using Resyone. The results show gaze analysis for the stages of getting out of bed using the equipment. One bar graph represents each step of getting out of bed, and the number of seconds it took is shown below. This number of seconds was taken as 100%, and the bar graph shows the percentage of time spent looking at each target or performing vocalization. Transiently looking at a face, fixation on a face, or vocalization were indicated using red, pink, and pale pink colors, respectively. When the gaze was looking at various parts of the subject’s body, they were shown in bluish colors. When the eye is looking at the bed or sheets, it is shown in a greenish color, and when looking at shoes or clothes, this is shown in a yellowish color.

**Figure 4 ijerph-21-00250-f004:**
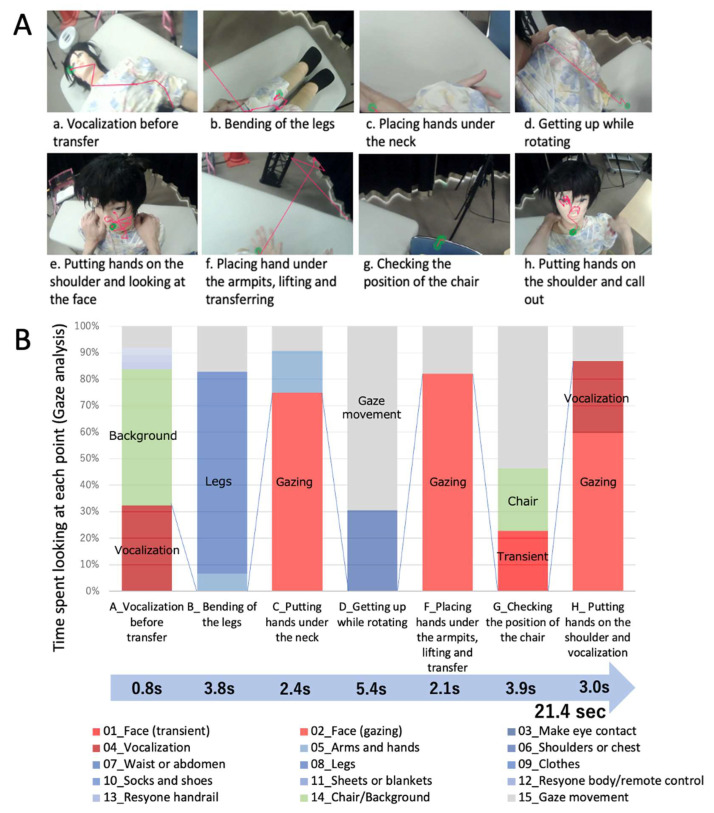
Analysis of a therapist’s gaze during a simulation in which a nursing practice doll is ‘helped’ to get up from the bed and move to a chair (without Resyone). The gaze in each step is shown in (**A**), and the gaze time for each step is shown in (**B**). The transfer steps are shown in the figure as vocalization before transfer (a), bending of the legs (b), placing hands under the neck (c), getting up while rotating (d), putting hands on the shoulders and looking at the face (e), placing hands under the armpits, lifting and transferring (f), checking the position of the chair (g), and putting hands on the shoulders and vocalization (h).

**Figure 5 ijerph-21-00250-f005:**
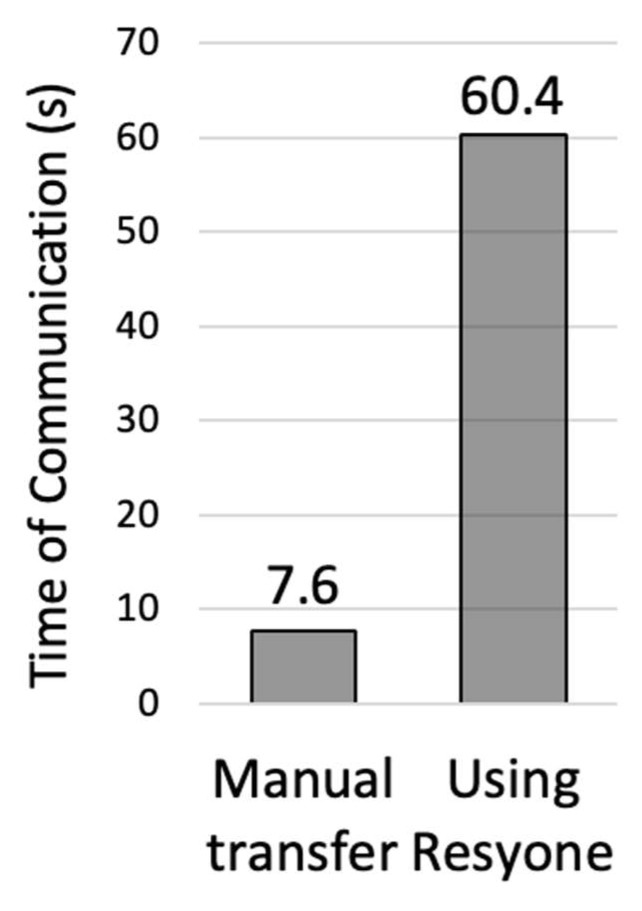
Time spent looking at the recipient’s face or vocalizing during transfer assistance.

## Data Availability

The datasets analyzed during the current study are not publicly available given that the research team has not completed its analysis, but they are available from the corresponding author on reasonable request with due consideration for the privacy of the subjects.

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
