# Peer review of "Robotic Care Equipment Improves Communication between Care Recipient and Caregiver in a Nursing Home as Revealed by Gaze Analysis: A Case Study"

_ijerph, 2024, doi:10.3390/ijerph21030250_

Round 1

Reviewer 1 Report

Comments and Suggestions for Authors

Dear authors

You approached an essential topic in the care of the patients: communication.

Your work is innovative and has good prospects for the future.

However, I have some doubts, comments and suggestions, that I think will improve the work:

1) If you studied only one person, why is your age in a range as well as the tale? It would be important to explain to the readers.

2) How did you obtain informed consent if the person cannot speak?

3) Your graphs are too challenging to read. Is there a chance to use better graphs or improve the explanation?

4) You performed a case study. I do not have any concerns or problems with that. But in the discussion, you should write about the limitations of this methodology and its impact on the results.

Please positively see my comments.

Kind regards

Author Response

We appreciated reviewers for their careful reading of the manuscript and helpful and thoughtful suggestions.
Here we have revised the manuscript as they mentioned. 

Point-by-point response

1) If you studied only one person, why is your age in a range as well as the tale? It would be important to explain to the readers.

 Response: We are sorry for this description. This is because the weight and age options in the questionnaire were set in a range of 5, and the participants checked those items. Details are now unknown as the subject is deceased. 

2) How did you obtain informed consent if the person cannot speak?

 Response: We have asked her husband to sign the informed consent on her behalf. So we added the explanation (P4, L137-8). 

3) Your graphs are too challenging to read. Is there a chance to use better graphs or improve the explanation?

 Response: Thank you for pointing it out. The explanation of the graph was insufficient. We have added more detailed explanation to figure legend (P4, L116-L122).

4) You performed a case study. I do not have any concerns or problems with that. But in the discussion, you should write about the limitations of this methodology and its impact on the results.

 Response: We have mentioned the limitations of this study and added impact on the results as "We have observed smiles on the subject's face, but it was not a quantitative result nor showing better quality of life. However, the current study covers a single type of robotic equipment and a limited number of subjects. It is necessary to consider facial expression analysis, etc. for further research." (P7, L228-31). 

Thank you, 
Best regards,

Reviewer 2 Report

Comments and Suggestions for Authors

The article "Robotic care equipment improves communication between care recipient and caregiver in a nursing home as revealed by gaze analysis: a case study" presents a case study highlighting the positive impact of using robotic nursing care equipment in a nursing home. The study was conducted in a nursing home where the bed-assisting robot "Resyone Plus" was employed.

The study concludes that the use of robotic nursing care equipment can enhance the quality of life for care recipients and reduce the burden on caregivers. However, the manuscript lacks mention of comparative data with other studies or control groups, and there is no comparison with similar equipment.

It is important to note that the entire study is based on the use of commercially available equipment. In other words, the efficiency of the device has already been tested for its intended purpose, making it challenging to identify the authors' contribution. The question arises: what sets this study apart? Is it in the usage, the definition of a unique protocol, or any statistical analyses?

Real-world case studies are always interesting, but it is crucial for authors to understand the expected contribution. Comparative data, well-defined experiment/usage protocols, and, in some cases, clinical parameters are necessary. A more detailed description of the sample's setting and conditions is warranted.

The mentioned equipment achieves its goal, but what is the differentiating factor? Is the aim to prove the equipment's effectiveness or showcase a particular technique? It is possible that I may not have identified it, so perhaps the authors need to be clearer and provide more detailed explanations of procedures, comparative data, and contribution.

Author Response

We appreciated reviewers for their careful reading of the manuscript and helpful and thoughtful suggestions.
Here we have revised the manuscript as they mentioned. 

Point-by-point response

The study concludes that the use of robotic nursing care equipment can enhance the quality of life for care recipients and reduce the burden on caregivers. However, the manuscript lacks mention of comparative data with other studies or control groups, and there is no comparison with similar equipment. 

 Response: We are sorry about limitations of the study as we have mentioned in Discussion (P7, L228-31). We could not set other subject who are not using robots. This is because original study protocol cannot follow the subject who are not using robots. We could expect that residents not using Resyone will not show improvements in facial expressions or quality of life. However, this should be the true control that reviewer mentioned. Also because the condition of the control subject cannot be expected to remain stable for even one year. In fact, three Resyones were introduced to three subjects, but two of them passed away within two years. The remaining one stopped using it, so the demonstration period was shortened. 
 Further, we compared our study with other studies (Refs 8, 18-22) in Dicussion (L223-225), but we could not find many studies. Further, it is not a true control that reviewer may be concerned, but we have performed a gaze analysis on a normal transfer as a kind of control (Figure 4). As for similar equipment, there are no similar equipment at all. This also the limitation of the study. We have added it as a limitation (L231-2). 

It is important to note that the entire study is based on the use of commercially available equipment. In other words, the efficiency of the device has already been tested for its intended purpose, making it challenging to identify the authors' contribution. The question arises: what sets this study apart? Is it in the usage, the definition of a unique protocol, or any statistical analyses?

 Response: Thank you for pointing it out. Nursing care devices like Resyone have been proven to be effective in reducing the burden of caregiving (intended purpose), but there has been no research at all regarding their effects on QOL. Prior to this study, we have investigated a study reported as Ref. 6. In the study, we have demonstrated the ability of additional care operations (new protocol in nursing care) to extend the life space of residents in a nursing home that was proficient in the use of Resyone. This study is one of the results obtained from a new protocol in care operations. We have mentioned those in Discussion section (L210-222).

Real-world case studies are always interesting, but it is crucial for authors to understand the expected contribution. Comparative data, well-defined experiment/usage protocols, and, in some cases, clinical parameters are necessary. A more detailed description of the sample's setting and conditions is warranted.

 Response: As mentioned above, this study is one of the result of a series of studies that we have performed during 2018 and 2020. We  have reported the effectiveness of device use in comunication between caregiver and the recipient and in extension of the life space (Refs. 4,5). We have mentioned the sample setting in Method section 2.2 Intervention period (L69-71). To be more specific, after 9 months of Resyone use on the subject (sustainably used), the caregivers were asked to apply additional care operations to extend the life space of the subject over a four-week period. This is our contribution to the care operation protocols in nursing homes. 

The mentioned equipment achieves its goal, but what is the differentiating factor? Is the aim to prove the equipment's effectiveness or showcase a particular technique? It is possible that I may not have identified it, so perhaps the authors need to be clearer and provide more detailed explanations of procedures, comparative data, and contribution.

 Response: We have added the differentiating factor (procedures and contribution, not comparative data) of our study to last paragraph of Discussion section (L233-238). 

Thank you very much for your concern,
Best wishes,